# The health benefits and cost-effectiveness of complete healthy vending

**Melda Lois Griffiths**[1]*, **Eryl Powell**[2], **Lucy Usher**[2], **Jacky Boivin**[1], **Lewis Bott**[1]

**1** School of Psychology, Cardiff University, Cardiff, Wales, United Kingdom, **2** Aneurin Bevan Gwent Public Health Team, Public Health Wales, Newport, Wales, United Kingdom

* GriffithsML4@cardiff.ac.uk

**Data Availability Statement:** The final dataset is available on the Open Science Framework, DOI 10.17605/OSF.IO/H2ZJ6.

**Funding:** MLG, LB, EP - Economic and Social Research Council grant ES/J500197/1. URL:

## Abstract

Vending machines contribute to growing levels of obesity. They typically contain energy dense, high fat snacks and attempts at persuading consumers to switch to healthier snacks sold within the same machine have had limited success. This study explored the health benefits and cost effectiveness of the complete replacement of regular snacks with healthy items. Two vending machines were manipulated in a 6-month trial, with a healthy and regular range of products alternated between the two machines every fortnight. Healthy vending resulted in a 61% drop in calories sold relative to regular vending, significant with time and product range as random factors. There was no evidence of compensatory behaviour from nearby shop sales nor in multi-item purchases from vending machines. The impact on profit was less clear. Sales dropped by 30% during healthy vending but variability across product range meant that the change was not significant. Overall our results demonstrate that complete healthy vending can be introduced in hospitals without a catastrophic loss in sales nor compensatory behaviours that offset the public health gains of consuming healthier products.

## Introduction

Vending machines are machines of convenience. Offering 24-hour access to food and drink in public spaces such as schools, hospitals and workspaces, they are many people's first choice when searching for a quick pick-me-up. It is therefore unsurprising that vending machines are predominantly stocked with quick-fix foods that are high in energy, but low in nutritional content. This has led some to describe vending machines as obesogenic food environments [1]. While the financial benefit of vending machines is undeniable, so too is their contribution to the obesity epidemic currently being evidenced across the globe [2]. With vending machines so widely dispersed across all manner of settings, addressing their contribution to the composition of the food environment is important for reducing obesity levels. In this study, we investigate one simple approach: the complete replacement of standard products typically considered unhealthy (i.e., high calorie, high fat products, such as chocolate bars) with healthier options. (i.e. products low in salt and added fat/sugar).

https://esrc.ukri.org/ The ESRC played no role in the study design, data collection and analysis, decision to publish, or preparation of the manuscript. LB - Aneurin Bevan University Health Board grant 512976. URL: https://abuhb.nhs. wales/ ABUHB played no role in the study design, analysis, decision to publish or preparation of this manuscript. The researchers were granted access to ABUHB premises to collect data.

**Competing interests:** The authors have declared that no competing interests exist.

Previous research has predominantly focused on environments in which healthy and regular products are both available and consumers have the choice of which to purchase. The goal has been to find ways of increasing the proportion of healthier choices made by consumers. We divide this literature into two general approaches. The first is to make healthier choices more appealing using labelling or promotional material. Two studies have shown some success in this respect. French et al. [3] applied health labels and health-promoting signage to 55 vending machines over 12 months. Low-fat sales accounted for 14.3% of all sales in the no label condition, but 15.4% with labels and signs. The increase was significant across vending location. In a school-based randomised control trial, Kocken et al. [4] increased the availability of healthier options and used hand signals to label products as either being favourable (thumbs up– 99 calories or less), moderately unfavourable (index finger– 100–170 calories) and unfavourable (thumbs down– 171 or more calories). The intervention led to moderately unfavourable products accounting for 26.6% more of all sales, and unfavourable products accounting for 27.6% less of all sales. Sales of favourable products were unchanged, with less than 2% of all sales coming from such items. On the whole however, promotional messages appear to have limited success. A review of the use of promotional materials and labels providing health information in vending machines [5], found that effects were either absent or minimal across the eight studies considered. It may well be that individuals do not read labels, and if they do, the majority may not find their messages to be sufficient motivation for changing food choice [6].

The second approach is to alter the relative costs of the choices, either by taxing regular products or by subsidising healthy products. Subsidising healthier products has a robust positive impact, with greater reductions in price leading to higher proportions of healthy sales [5]. For example, French et al. [3], described above, applied price reductions of 10%, 25% and 50% on low-fat products, and observed significant increases in sales of healthy products of 1.0%, 4.2%, and 10.1% respectively, and with no loss in revenue. Taxing unhealthy options has been tested far less extensively but existing evidence shows similar benefits to subsidising healthy options. A 25% tax on unhealthy items was tested in a 14-month trial conducted across three vending locations [7]. In addition to the tax, the researchers tested a 25% discount on healthy items, a 25 second delay in the vending time of unhealthy options and a combination of time delays and pricing interventions. All interventions successfully increased the proportion of healthy choices made and the most effective was the 25% price tax on unhealthy products, which increased the proportion of healthy choices made within the machine by 13.6%.

Substantial health gains can be made by manipulating price and other factors within mixed vending machines [3, 5, 7]. Nonetheless, more impressive changes in health behaviour might be obtained by removing regular products entirely and leaving only healthy products. If people have no choice about choosing healthy products, they will surely become healthier. Prior studies adopting the complete replacement approach are few in number but those that exist suggest some difficulties. The first is that consumers may go elsewhere to search for unhealthy products. While vending machines follow certain healthy eating restrictions, other vendors do not. If consumers are frustrated or dissatisfied with the offering in the machine, they may search for alternatives at other nearby outlets. While vending would be 100% healthy, behaviour would not necessarily be so. This problem is illustrated by studies examining the effect of removing sugared soft drinks from vending machines in schools. Taber, Chriqui, Frank and Chaloupka [8] measured the association between vending access to soft drinks and consumption in 10,000 schools across the United States. They found that overall consumption of soft drinks was higher in children where vending of soft drinks was prohibited. Taber and colleagues [8] suggested that without a whole-environment change to soft drink supply, children were able to obtain drinks elsewhere, where they overcompensated for the removal of soft drinks from the school vending machines. In the same vein, evidence from another school

study shows that while soft drink consumption on school premises decreased where vending machines did not sell soft drinks, overall consumption (including off premises) did not differ across groups [9]. This highlights the need to evaluate the effectiveness of vending interventions not only based on what happens inside the machine, but also on what happens outside of it (c.f. [3, 7, 10]).

A second concern for interventions that completely remove unhealthy products is that consumers may buy more healthy products to replace the perceived loss of satiation that occurs from not consuming unhealthy products (e.g. buying two healthy bars to replace one unhealthy bar). There is no evidence for this compensatory behaviour but nor are there any vending studies that have tracked the behaviour of individual consumers in real-choice scenarios. Data showing an increase in the sales of healthy products (e.g. with price reductions in French et al. [3]) are consistent with fewer individuals making more multi-item purchases. Furthermore, the advent of credit card readers in vending machines removes some of the impediments to multi-item purchasing (e.g. less need for coinage, less time needed per transactions), and so multi-item purchases might be more frequent now than in the past. If multi-item purchasing is more common in machines that stock only healthy items, behaviour may not necessarily be healthier.

A final concern with complete healthy vending is that the loss in revenue may be unsustainable. Balancing cost and public health benefit is a common concern for health interventions [11, 12]. Catering departments may be reliant on vending machine revenue, and any action that would jeopardize these profits may be approached with reluctance. Previous vending studies have not observed a drop in profit during healthy vending interventions (e.g. [3, 4, 7]). However, the interpretation and relevance of these studies with respect to completely healthy vending is complicated. First, previous studies have all used partial healthy vending. Particularly determined consumers have always had the option of choosing their regular products if they wished. As noted, if regular products are not available, consumers may switch to purchasing elsewhere or to purchasing nothing at all. Complete healthy vending would suffer a drop in revenue from these consumers whereas partial healthy vending would not. Second, some partial healthy vending interventions have included salient price discounts on healthy products [3, 4, 7]. The discount has the effect of increasing sales volume by creating extra demand, thereby offsetting any drop in demand for the regular product (and increasing the total number of calories consumed). In complete healthy vending, any discount applied to healthy products would not be salient because there would be no non-discounted products with which to compare it. Discounting healthy products within complete healthy vending may not be a viable option for balancing revenue. Finally, the product range used across studies varies considerably and random effects of product range have not been included in statistical analyses. While these studies have demonstrated that their interventions are both health and cost effective within their product ranges, it is difficult to know whether effects on sales volume (or any other dependent measure) would replicate in environments with different product ranges.

In summary, complete replacement of unhealthy products may have the potential for large health improvements. Nonetheless, it may result in problematic compensatory behaviours, and implications on revenue are unknown. In this study we report a controlled experiment to test the health and cost implications of total healthy vending. Sales data was collected from vending machines located in a hospital environment. Stock was varied fortnightly between a regular (unhealthy) range and a healthy range over the course of six months. To assess healthy behaviour, we measured calorific content of products sold, and to measure sales, we analysed sales volume and cost/profit. The analysis for both used linear mixed models with time and product range as random effects. Potential compensatory behaviours were also explored. To establish whether consumers would search elsewhere for regular snacks during healthy

vending periods, we analysed sales data from the nearby convenience shop, with the hypothesis that unhealthy sales would increase at the shop during healthy vending. Finally, we used credit card data and on-site observations to test the hypothesis that individuals would compensate for the lack of unhealthy options available at the machine by making more multi-item purchases during healthy vending periods than regular periods.

## Materials and methods

### Participants

Participants voluntarily made purchases from vending machines without being aware that they were taking part in a study. Ethical approval was granted from the NHS research ethics committee (IRAS number: 231390) and the Cardiff University School of Psychology ethics committee. Permission was given not to collect consent from participants and to not inform them of the nature of the study.

### Study design

The independent variable was vending condition (two levels: healthy vending, unhealthy vending). Healthy and unhealthy stock was alternated across two vending locations within a large hospital site. Healthy stock was available in one vending location and, concurrently, unhealthy stock in the other. Stock type was systematically rotated across the two locations every two weeks. The study lasted for 24 weeks. Across the duration of the study, each vending machine contained healthy stock for 12 two-week periods and unhealthy stock for 12 two-week periods. Sales data from the vending machines were used to measure three dependent variables—sales volume, profit and calories sold.

### Implementation

Vending machines were located in public areas, where staff, patients and visitors could all access the machines. One machine was located in an Accident and Emergency department (*A&E* machine), the other in a reception area (*reception* machine). The A&E machine was in an isolated location, separated from other food outlets. The reception machine was located within the main entrance for maternity patients, and on the same floor as a coffee shop and restaurant, which sold a variety of snacks and hot and cold meals (see S1 Appendix for hospital floor plan).

Vending machines were identical (Model: Necta Tango) running the Nayax vending system. The machines allowed cash and card purchases. They had 32 coils, 16 for crisps (four items in each of the four top rows) and 16 for bars/small packets (eight items in each of the bottom two rows). Product sold, location, exact time and method of purchase (cash or card; first and last four digits recorded if card) were all recorded by the Nayax system.

Planograms (diagrams of the planned visual layout of stock) were kept constant throughout the trial (see S2 Appendix). The fidelity of the compliance to this layout was monitored by supervising the first three changeover periods (when stock from each machine was swapped over) and by receiving timestamped photographic evidence of the layout from each changeover thereafter. Time estimates for each changeover period during the trial were also obtained from the vending machine providers, to ensure that these were kept consistent and could be cross-referenced with the sales data to ensure all sales were logged under the correct condition.

## Vending snacks

Snacks were selected by the vending provider (JDJ Vending Services) and the research team (including a Public Health Wales Consultant and a Public Health Wales Dietitian). All snacks were selected with the aim of maximizing profit subject to the healthy/unhealthy experimental constraints.

Healthy snacks satisfied the Welsh Hospital Healthy Vending directive constraints [13], whereas unhealthy items did not. The government-developed constraints set strict guidelines for fat, saturated fat, sugar and salt levels for all vending products (see S5 Appendix for further details). For the healthy condition, there were 19 distinct snacks, 13 of which occupied two coils, and for the unhealthy condition, there were 23 distinct snacks, 9 of which occupied two coils. The size of the range differed across conditions because more unhealthy snacks were available from the vendor than healthy snacks. Details of the nutritional information for each individual snack can be seen in S3 Appendix. The mean wholesale cost of products was comparable across healthy and unhealthy products, $M_{\text{Healthy}}$ = 44p (SD = 5p), $M_{\text{Unhealthy}}$ = 43p (SD = 3p), and to ensure that products were equally affordable for all potential customers, all products were sold for 80p. Wholesale cost and profit margins for each product can be found in S4 Appendix.

## Observational study

An observational study was completed concurrently with collection of the vending data. The goal was to obtain information about multiple purchases. Observations occurred during two time points of the study–a five-day period in the first changeover period, and a five-day period in the sixth changeover period. Observations were made Monday to Friday, from 11:30am to 17:30am. The researcher alternated between observing each machine every 1.5 hours, with the order of observation being counterbalanced across the five days. During the observation period, data was collected on the number of items purchased by each individual consumer, and the time it took to complete their purchasing (starting from when they approached the machine, and the last time measure recorded being that noted for the collection of the last item purchased). The researcher sat in a location where the machine was in view and recorded all purchases on a mobile device. As both machines were located in areas where members of the public often sat waiting, this was regarded an appropriately covert method.

## Shop comparison

The reception vending machine was located close to a shop selling healthy and unhealthy snacks. The shop was open 8am to 4pm, seven days a week, and was approximately 25 meters from the vending machine. To establish whether healthy snacks in the vending machines led to replacement purchasing of unhealthy snacks elsewhere, we obtained sales data for unhealthy shop products during the study period. See Results for analysis.

## Statistical analyses

The data were analysed as mixed models with the lme4 package in R [14]. Individual products (*items*) and time period were included as random effects. Model specification was maximal (see [15]), in that all possible random effects parameters were included. *p*-values were computed with the Kenward-Roger and Satterthwaite approximations to degrees of freedom, implemented in the lmerTest package [16].

## Results

Across the 6-month trial period, a total of 17,571 sales were made across both machines. Of these, 9959 sales were made in A&E and 7612 sales were made in the reception area. The sales data from both machines in both conditions of the experiment were used to explore the impact on the healthiness of behaviour and the cost of the intervention.

### Health behaviour

The calories sold across the healthy and unhealthy conditions of the experiment were compared. The healthy range of items had a lower mean calorie content per item than that of the unhealthy range, $M$ = 130.50 kcals (SD = 41.46) *vs M* = 226.63 kcals (SD = 65.99). Nutritional information for each item on sale was mapped onto sales volume data within each condition of the experiment. 923,000 kcals were purchased in the healthy condition and 2,354,000 kcals were purchased in the unhealthy condition. This equates to a 61% drop in the number of calories sold from the unhealthy condition to the healthy condition.

To assess whether the drop in calories sold was robust across time and item range, we obtained an average calorie sold score for each fortnightly time period and tested a mixed effects model with product condition (healthy, unhealthy) and machine (A&E, reception) as fixed effects, and fortnightly time period (N = 12) and item range (healthy N = 19; unhealthy N = 23) as random effects. Random intercepts and slopes were included for time period and random intercepts for items.

Significantly fewer calories were sold in the healthy product conditions of the experiment, β = -2240.74; SE = 579.23; t = -3.868; $p$ < 0.0001 (see Fig 1). In addition, significantly more calories were sold in the A&E machine than the reception machine, β = 742.25; SE = 184.38; t = 4.026; $p$ < 0.0001, but there was no interaction between them, β = 41.86; SE = 208.85; t = 0.2; $p$ = 0.84. In short, stocking healthy items successfully lowered the number of calories sold relative to stocking unhealthy items.

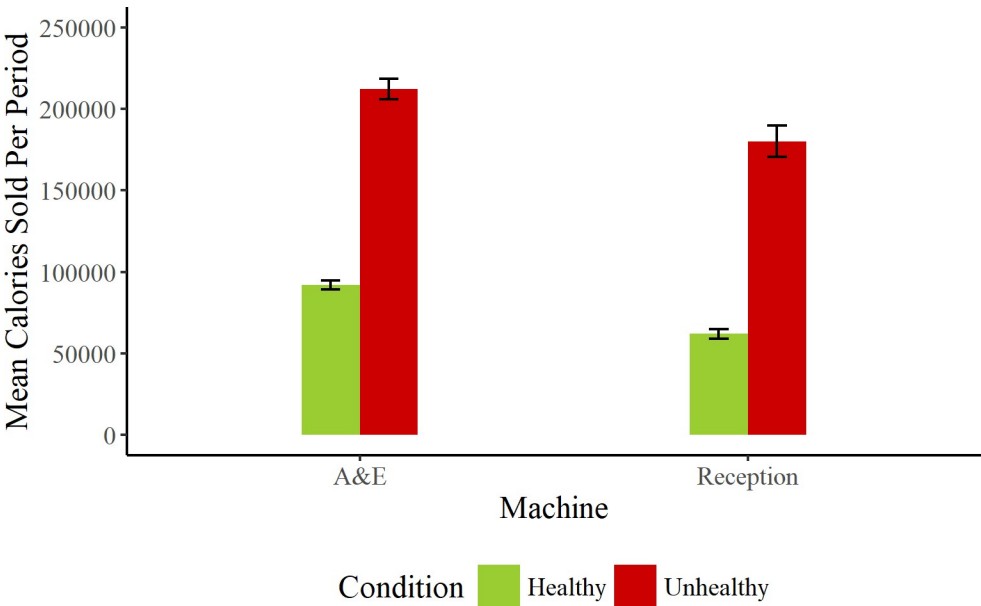

**Fig 1. Mean number of calories sold within a fortnightly period for each product type, in each machine.** Error bars show standard error of the mean with respect to time period.

## Sales volume

Sales were higher in the unhealthy condition than the healthy condition, N = 10,155 unhealthy and N = 7,416 healthy sales being made across the 6 months of testing. In a similar approach to the assessment of calories, we calculated sales volume for each of the 12 periods of the experiment and tested a mixed effects model with period and item as random effects (see Fig 2). Random intercepts and slopes were included for time period and random intercepts for items.

Surprisingly, we observed no significant effect of product condition, $\beta$ = -2.134; SE = 2.818; t = 0.453; $p$ = 0.45, nor of the interaction of machine by product condition, $\beta$ = 1.523; SE = 1.096; t = 1.389; $p$ = 0.18. Sales volumes did however differ across machines, with a significantly higher number of sales being recorded in A&E compared to the reception area, $\beta$ = 4.802; SE = 0.915; t = 5.248; $p < 0.0001$.

To better understand why there were no significant differences across product conditions, we examined the data in more detail. Figs 3 and 4 show the data as a function of time and machine.

For both machines, there were more sales for unhealthy products than healthy products across the entire time span of the experiment. To test this, we constructed a model that included product range and machine as fixed factors and time as a random factor, but not item. This model highlighted significantly greater sales in the unhealthy product condition, $\beta$ = -2.134; SE = 0.880; t = -2.425; $p$ = 0.016. There was no interaction of product condition by machine, $\beta$ = 1.523; SE = 0.880; t = 1.731; $p$ = 0.08.

This pattern of results (significance of product type dependent on inclusion of item as random effect) suggests that across item variability for sales volume was high. Fig 5 illustrates this graphically.

While there were more sales on average in the unhealthy condition, both product conditions have high variability, with some products selling as much as 80 units in each group, and some as few as 10 units. The high variability ultimately prevents the model from showing

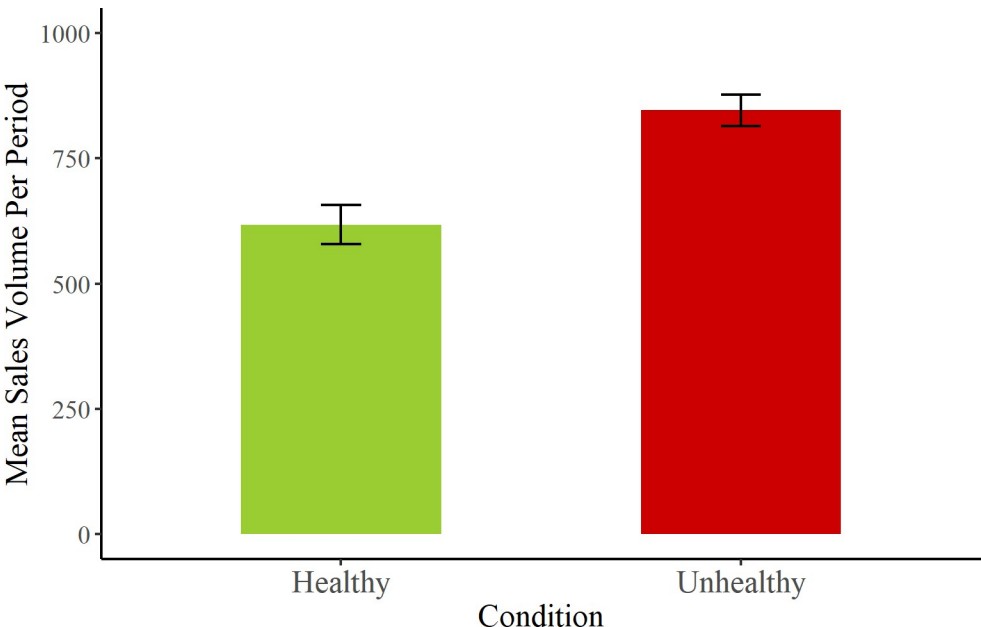

**Fig 2. Mean sales volume per period within each condition.** Error bars show standard error of the mean.

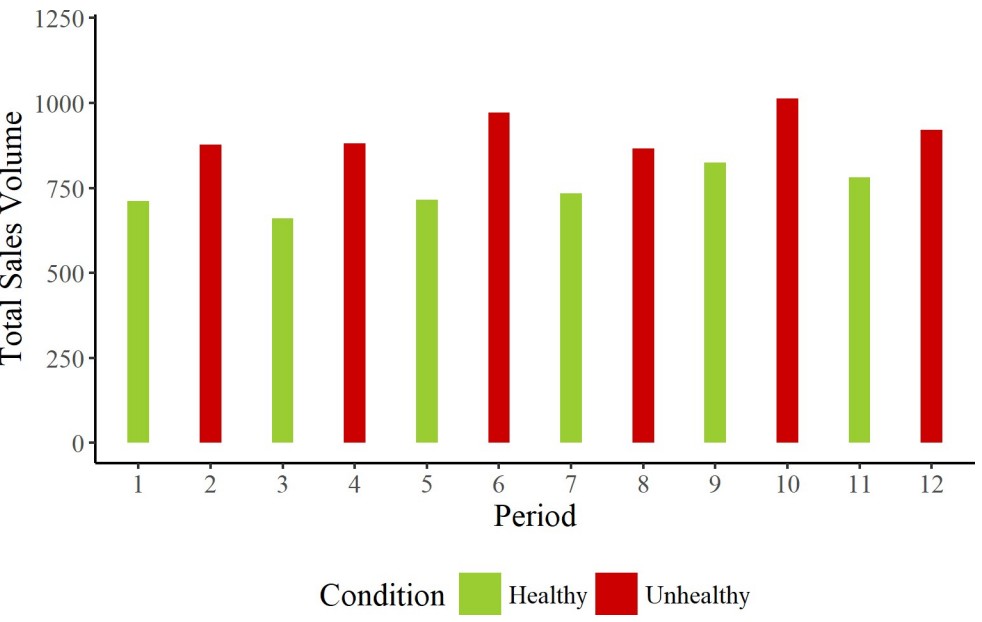

**Fig 3. Total sales volume in the A&E machine during each of the 12 fortnightly periods.**

significant effects of the different product ranges on sales. We consider the meaning of the result further in the General Discussion.

## Profit

We defined profit as the difference between cost price and sale price. All products were sold at the same price, 80p. Healthy and unhealthy ranges yielded similar profits per item, $M = 41p$ (SD = 3) $vs$ $M = 41p$ (SD = 5) respectively. Thus the difference in total profit across conditions

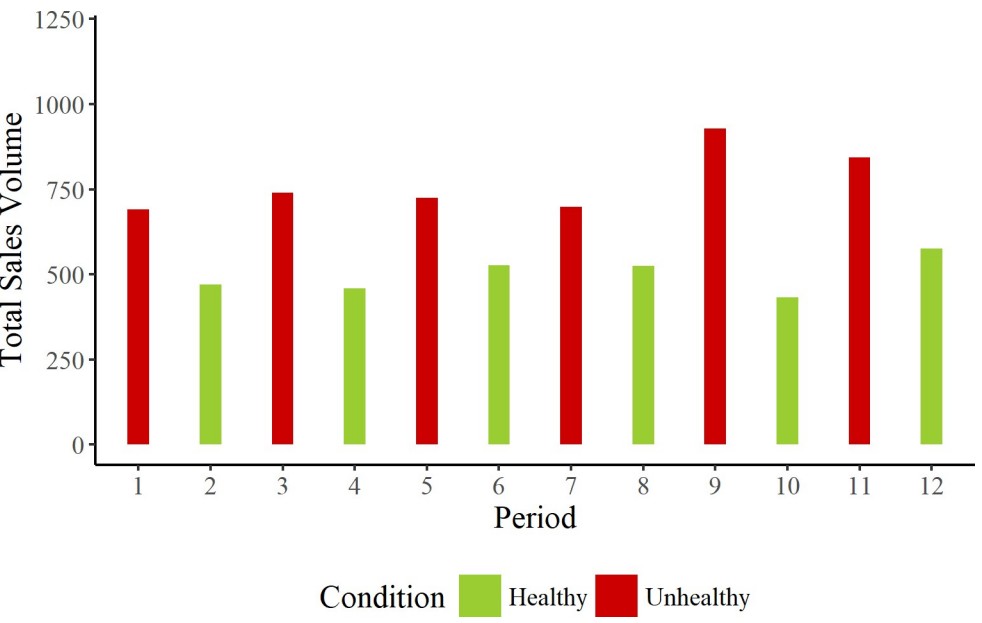

**Fig 4. Total sales volume in the reception machine during each of the 12 fortnightly periods.**

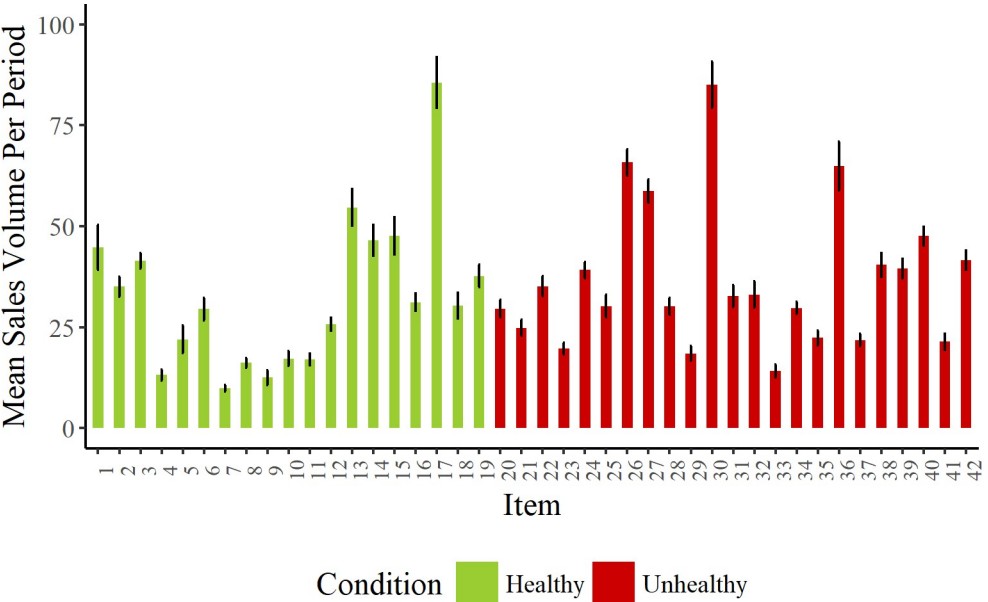

**Fig 5. Mean sales volume per period for each item (grouped by condition).** Each bar represents a different item (numbers for each item correspond to those in S3 Appendix). Error bars show standard error of the mean.

was largely determined by the number of sales in each condition. Total profit was £2657.70 in the healthy condition, and £3773.86 in the unhealthy condition.

Not surprisingly, inferential statistics for profit yielded similar conclusions to those of total sales. A mixed model with profit as the dependent measure, machine and product condition as fixed factors, and time and item as random effects revealed no significant effect of product condition, $\beta$ = -1.008; SE = 1.102; t = -0.915; $p$ = 0.37, but the same model without items as a random effect produced a significant result, $\beta$ = -1.008; SE = 0.340; t = -2.962; $p$ = 0.0032. The effect of machine was significant in both analyses, (analysis including item as random effect: $\beta$ = 1.757; SE = 0.334; t = 5.265; $p$ < 0.0001; analysis excluding item as random effect: $\beta$ = 1.757; SE = 0.340; t = 5.160; $p$ < 0.0001), but there was no interaction of machine by product range in either analysis, (analysis including item as random effect: $\beta$ = 0.497; SE = 0.396, t = 1.2455; $p$ = 0.22; analysis excluding item as random effect: $\beta$ = 0.497; SE = 0.340; t = 1.460; $p$ = 0.14).

### Calorie content and sales

The majority of healthy sales came from products under the 100-calorie mark (53%), all bar one of these being crisps. None of the unhealthy products were below this 100-calorie threshold. Within both the healthy and unhealthy product ranges, there was no significant association between the calorie content of products and their total sales volumes, (Healthy range: r = -0.291, n = 19, p = 0.23; Unhealthy range: r = -0.140, n = 23, p = 0.53).

### Shop sales

To establish whether participants engaged in compensatory purchasing elsewhere we examined the sales data of the convenience shop located approximately 25m from the reception vending machine. Items sold in the shop were categorized as either healthy or unhealthy, and the total sales volumes for unhealthy items were calculated across the twelve fortnightly periods of the experiment. If people purchased unhealthy products from the shop instead of healthy products from the vending machine, sales of unhealthy products from the shop should

be higher when the vending machine sold healthy products than when they it sold unhealthy products.

We conducted the analysis on two sets of products, a restricted range of shop products comparable to that sold in the vending machine (N = 17; the remaining 6 were not sold by the shop), and the entire set of unhealthy products sold in the shop (N = 84). Items were coded as unhealthy/healthy based on the same Welsh CMO guidelines used to categorize vending machine items. Both analyses revealed almost identical fortnightly sales across conditions. For comparable products, healthy vending $M = 842$ unhealthy shop sales ($SD = 91$) $vs$ unhealthy vending $M = 843$ unhealthy shop sales ($SD = 118$); and for all products, healthy vending $M = 2757$ unhealthy shop sales ($SD = 202$) $vs$ healthy vending condition $M = 2721$ unhealthy shop sales ($SD = 322$). A mixed model with sales of the complete shop range as the dependent measure, product condition (healthy, unhealthy) as a fixed factor, and time and item as random factors (random slopes and intercepts for product range) did not show a significant effect of product condition, $\beta = 0.211$; SE = 0.848; t = 0.249; $p = 0.81$, nor did a similar model applied to the comparable product range, $\beta = -0.010$; SE = 1.964; t = -0.005; $p = 0.99$ (similar qualitative results were obtained when item was excluded as a random factor, see Sales Volume analysis). This suggests that participants were not shifting their purchasing from the vending machine to the shop when the vending machine sold healthy products.

## Estimates of multi-item purchasing

We also tested whether individuals engaged in compensatory behaviours by making more multi-item purchases when approaching the machine during healthy vending compared to unhealthy vending. Two estimates of multi-item purchasing were obtained. The first was from an observational session and the second from a tally of purchases made on the same credit cards.

**Observation.** A tally was kept of the number of items bought by each individual who approached the machine during observation. One hundred and nine individuals were observed making single, double and triple item purchases. However, there was no significant difference in the likelihood of single item $vs$ multi-item purchases across product conditions (see Fig 6), $\chi^2 (1) = 2.20$, $p = 0.14$.

**Credit card analysis.** The vending machine provided the first and last four digits of each credit card that made a purchase. Purchases made by cards with the same first and last four digits were treated as multiple purchases made by the same consumer.

For those that paid with a credit card, multi-item purchases appeared to be more common than single-item purchases for both the healthy and unhealthy product ranges, with this pattern more pronounced for unhealthy products (see Fig 7). However, there was no significant difference in the likelihood of single item $vs$ multi-item purchases across product conditions, based on credit card sales, $\chi^2 (1) = 1.41$, $p = 0.23$.

## General discussion

The aim of this study was to test the impact on health behaviour and cost effectiveness of replacing unhealthy products with healthy products in vending machines. For health, the data is unambiguous. Healthy vending led to 61% fewer calories being sold. Furthermore, the drop in calories sold was not associated with increased sales of unhealthy products in a nearby food outlet, nor did it result in more multi-purchase sales from individuals seeking to replace lost calories from healthy products. These effects were significant across time and product range. For cost, the effects were less clear. While there was a 27% drop in sales volume and a 30% drop in profit associated with healthy vending, these effects were not significant. Overall our

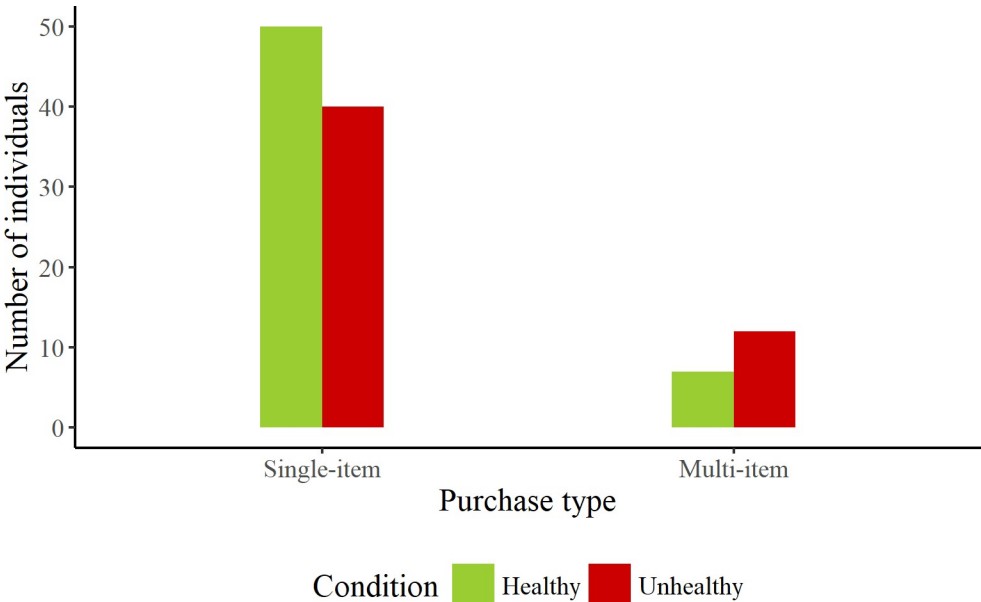

**Fig 6. Number of consumers making single or multi-item purchases in healthy and unhealthy conditions during observation.**

results demonstrate that complete healthy vending can be introduced in hospitals without incurring a catastrophic loss in sales nor compensatory behaviours that offset the public health gains of healthier products.

Healthy behaviour was achieved when removing all unhealthy products from vending machines. Previous research has yielded much lower health benefit. Products under 100 kcals made up 53% of all healthy sales in the present study. In contrast, Kocken et al. [4] were unable

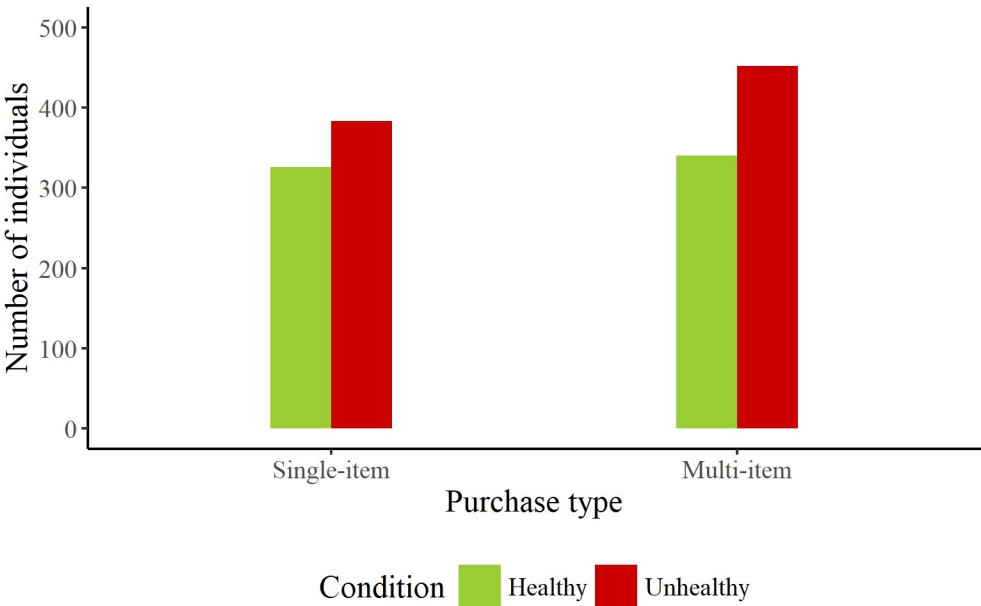

**Fig 7. Number of consumers making single or multi-item purchases in healthy and unhealthy conditions with credit cards.**

to increase the proportion of sales within this category above 2% in their school-based study where healthier products were made more available, more apparent (through labelling) and more affordable. By using diverse initiatives for healthy products, healthy sales as a proportion of all sales can be achieved: 21% with a 50% price reduction [3], 45% with a 75% healthy vending intervention [4], 53.7% with a 25% tax on unhealthy products [7], and 78% by restricting unhealthy products to 33% of the assortment [10]. In the present study, complete healthy vending by definition led to 100% healthy purchasing. Clearly, if the primary aim of an intervention is to increase healthy behaviour, completely replacing regular products with healthy products will have the greatest impact.

What is surprising about our intervention is that it did not lead to a dramatic loss in sales or profit. On average, sales in the healthy condition were 73% of those in the unhealthy condition (profit 70%), and this change was statistically significant only in a secondary analysis when the random effects of items were removed from the statistical model (see Implications for vendors, below). That healthy products resulted in a high proportion of sales is important for maintaining consumer satisfaction and for encouraging vendors and hospitals to maintain healthy products.

In addition, there was no evidence that consumers engaged in compensatory behaviours that offset the benefits of healthy vending. We tested two sorts. The first relates to purchasing at the individual level. Analysing averages, as has been done in previous vending research, ignores information about how behaviour manifests itself at the individual level. Average behaviour may improve after an intervention but health improvements in some groups may obscure detriments in others. In our study we were able to monitor individual purchasing by analysing single and multi-purchase behaviour. We were concerned that some groups may have stopped consuming entirely while others may have chosen to make more multi-purchases to offset the lower perceived satiation of the healthy options. Contrary to this hypothesis, however, we found uniform purchasing behaviour across healthy and regular conditions, with the vast majority being single-item purchases. From a public health perspective these findings are encouraging.

Measures of the second compensatory behaviour were similarly positive. Previous research has suggested that imposing bans within certain environments can lead to replacement behaviour occurring elsewhere [8, 9]. Here, regardless of what was being sold in the reception vending machine, sales of unhealthy items at the vendor 25m away remained stable. Consumers did not seek out the closest available alternative when faced with a healthy selection of products, which in turn suggests that vending machine custom is to an extent captive custom. Consumers will buy at the machine, or if unsatisfied with what's on offer, they will not buy anything at all. Both of these outcomes have a positive health impact (compared to regular vending). However, we acknowledge that there are other compensatory behaviours that our research did not capture, e.g. whether individuals brought more snacks from home during healthy vending, and that other methods, e.g. interviews with consumers, are required to gain a complete understanding of what consumers do when faced with unexpected food choices.

## Implications for vendors

Profits and sales were first analysed by including time and product range as random factors in a mixed model. Although there was a 27% drop in sales (30% drop in profit) with the introduction of healthy products, the effect was not significant. When the random effect of product range was removed, however, the drops in sales volume (27%) and profit (30%) were significant (across time). This means that if the same products were used in the future, sales volumes/profits from healthy products would be significantly less than those from unhealthy

products, but if different products were used, the pattern may not be the same. Whether organisations use a similar range of products as used in the present study depends on the similarity of the vending environment. For example, hospitals in Wales will have access to the same kinds of products as we used in our study and so are likely to experience a similar drop in profit when switching from unhealthy to healthy vending, but those in other countries will have access to a different range of products and should expect a different pattern of sales.

From a vendor's perspective the implication is that the product range is crucial in determining the size of the sales change from unhealthy to healthy vending. A selection of the healthy products in our range sold very well, with the highest selling product across both conditions being a healthy one (see Fig 5). The healthiness of a product is therefore not necessarily a predictor of its saleability. While our product range produced losses at the machine level, exploring sales item by item can help determine which products might have potential for making healthy vending more profitable.

## Changing consumer behaviour vs changing consumers

The change from unhealthy to healthy products could have been accompanied by two sorts of behavioural patterns. One possibility is that there was one specific type of consumer whose behaviour was changed as a consequence of the intervention–those with a preference for unhealthy products. In the healthy condition, such consumers could either be flexible and choose to purchase a healthier product instead, or if they were more inflexible and unsatisfied with the selection, refrain from making a purchase at all (accounting for the difference in sales seen across conditions). The second possibility is that purchases in each condition were made by two separate types of consumer—those that had a preference for unhealthy products, and those had a preference for healthy products. The unhealthy products attracted only the unhealthy consumers and the healthy products only the healthy consumers. With this explanation, our findings would suggest that the unhealthy group were more numerous than the healthy group.

Categorically distinguishing between these accounts is not possible with our data. An additional, mixed condition would be required in which the entire range of healthy and unhealthy products would be available in a single machine. Comparison between healthy and mixed conditions would indicate the proportion of consumers who changed their preference over and above those who were attracted to the healthy range. However, in our view, the two consumer model is unlikely. This account would require that in the healthy condition, a large number of consumers would have approached the vending machine only to leave empty-handed when they found that no unhealthy products were available. Yet during the observation phase of the study we did not notice such behaviour (although we did not explicitly code for customers not making purchases). Furthermore, if consumers were sufficiently focussed on obtaining an unhealthy snack, they would likely have searched out an alternative outlet that would provide one. Even though the closest outlet to the reception machine was only 25m away, we observed no evidence of such compensatory behaviour.

## Limitations and future directions

There are some limitations to the study. The first is that we tested in only a single location (a hospital). It is possible that different locations would yield different patterns of purchasing and compensatory behaviour. For example, hospitals are environments in which healthy behaviour is to be expected and some consumers might approach vending machines with the intention of purchasing healthy snacks. Consequently, the drop in sales with healthy products reported here might be smaller than in other environments in which regular products are expected.

Similarly, hospitals have fewer opportunities for alternative venue purchases than other environments such as airports. Compensatory behaviour may be greater where there are more outlets closer to the vending machines. Furthermore, the product guidelines we adopted were developed specifically for hospitals. Our findings may not translate to environments with alternative vending guidelines. For example, guidelines in Welsh schools are arguably more exclusionary, with most confectionary and savoury snacks banned within them [17]. The impact on sales or the prevalence of compensatory behaviours may be greater if replicating the study in a school environment. Nevertheless, our findings are still a useful reference point for what could be applied to environments that are currently free of any vending restrictions. Another limitation is the failure to account for stock waste in our estimate of the cost of the intervention. Decreases in sales may result in increases in waste due to products passing their shelf-life without selling. Future work should measure not only how many products are sold, but how many products are not sold, for a more accurate representation of the cost of healthy vending. Finally, we did not assess overall satisfaction levels for the product range tested. Complete replacement is an intrusive approach. High intrusiveness can put the longevity of interventions at risk, as they are more likely to cause frustration and dissatisfaction within those that are subjected to them [18].

## Conclusions

In conclusion, this study provides evidence about the health benefits and cost effectiveness of complete healthy vending. Under healthy vending, we saw a substantial drop in calories consumed but a surprisingly small drop in sales and no noticeable increase in compensatory behaviours. Consumers appear to purchase what is available to them, even if it is a healthy product, rather than going elsewhere. This suggests that healthy vending is an effective health improvement intervention for hospitals that has few negative consequences.

## Supporting information

**S1 Appendix. Hospital floor plan.**
(DOCX)

**S2 Appendix. Vending machine layout.**
(DOCX)

**S3 Appendix. Nutritional information.**
(DOCX)

**S4 Appendix. Product cost and profitability.**
(DOCX)

**S5 Appendix. Nutritional guidelines for vending.**
(DOCX)

## Acknowledgments

We wish to thank the staff of JDJ Vending Services for their support and cooperation in conducting this study. In addition, we are grateful to the Aneurin Bevan University Health Board Facilities team, Arabella Roberts and Gareth Hughes, for supporting us in bringing this study through from conception to completion.

## Author Contributions

**Conceptualization:** Melda Lois Griffiths, Eryl Powell, Jacky Boivin, Lewis Bott.

**Data curation:** Melda Lois Griffiths, Lewis Bott.

**Formal analysis:** Melda Lois Griffiths, Lewis Bott.

**Funding acquisition:** Eryl Powell, Lewis Bott.

**Investigation:** Melda Lois Griffiths, Lewis Bott.

**Methodology:** Melda Lois Griffiths, Lucy Usher, Jacky Boivin, Lewis Bott.

**Project administration:** Melda Lois Griffiths, Lewis Bott.

**Resources:** Melda Lois Griffiths, Eryl Powell, Lucy Usher, Lewis Bott.

**Software:** Melda Lois Griffiths, Lewis Bott.

**Supervision:** Eryl Powell, Jacky Boivin, Lewis Bott.

**Validation:** Melda Lois Griffiths, Jacky Boivin, Lewis Bott.

**Visualization:** Melda Lois Griffiths, Lewis Bott.

**Writing – original draft:** Melda Lois Griffiths, Lewis Bott.

**Writing – review & editing:** Melda Lois Griffiths, Lucy Usher, Jacky Boivin, Lewis Bott.

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
