## [Decision Letter · Decision Letter 0]

2 Jul 2020

PONE-D-20-11945

The health benefits and cost-effectiveness of complete healthy vending

PLOS ONE

Dear Dr. Griffiths,

Thank you for submitting your manuscript to PLOS ONE. After careful consideration, we feel that it has merit but does not fully meet PLOS ONE’s publication criteria as it currently stands. Therefore, we invite you to submit a revised version of the manuscript that addresses the points raised during the review process.

We look forward to receiving your revised manuscript.

Kind regards,

Zhifeng Gao

Academic Editor

PLOS ONE

Journal Requirements:

3. Please ensure that you refer to Figure 4 in your text as, if accepted, production will need this reference to link the reader to the figure.

Additional Editor Comments (if provided):

Reviewers' comments:

Reviewer's Responses to Questions

**Comments to the Author**

1. Is the manuscript technically sound, and do the data support the conclusions?

Reviewer #1: Yes

Reviewer #2: Yes

2. Has the statistical analysis been performed appropriately and rigorously? 

Reviewer #1: Yes

Reviewer #2: I Don't Know

3. Have the authors made all data underlying the findings in their manuscript fully available?

Reviewer #1: No

Reviewer #2: Yes

4. Is the manuscript presented in an intelligible fashion and written in standard English?

Reviewer #1: Yes

Reviewer #2: Yes

5. Review Comments to the Author

Reviewer #1: The paper describes an elegant experimental design conducted in real life (hospital) through two identical vending machines of which over a six month (24 weeks) period the assortment is systematically varied in two week periods, alternating between 19 healthy snacks (13 of which in double coils and 23 unhealthy snacks (of which 9 occupied two coils). Choices are being recorded in terms of sales volumes (17571 sales made 10155 from the unhealthy and 7416 from the healthy assortments), representing 2,354,000 and 923,000 kcals respectively.

Additional information is collected to explore potential compensation effects reflected in multi-item purchases per shopping trip and compensation in nearby shop sales. Sales volumes were translated into profit estimates on the basis of margin (sales price - cost price); as margin is quite similar for healthy and unhealthy options this holds a (almost) linear relationship to changes in sales volumes.

Bottom line the paper seems to conclude: (1) that sales volumes drop by 27%, representing a 61% drop in calories purchased when the healthy assortment is made available rather than the unhealthy, (2) that there is a 30% drop in profit (2657.70 vs. 3773.86 BP) but that this was not significant because of large variation across items in the assortment, and (3) that there are no compensation mechanisms identified (multi item purchases and near shop purchases).

Although the experimental design in real life is really an attractive feature of this research, and the results are generally convincing, there are some critical issues to be reflected on. Each of those will impact on the discussion section, which in my view requires more work:

1. What do the results actually show us? In other words what is the substantive contribution to the field?

Does not the paper essentially show that if (mildly) hungry people face an assortment they will select an option that best fits their preference structure (or refrain from buying if it is not there). This, by definition, will lead to fewer and lower caloric choices in the healthy assortment condition. This is because, assuming that preferences are, on average, biased towards less healthy options, it is less likely that a preferred option is available and if it is, in the healthy assortment, it will by definition lead to reduced caloric intake.

It would be worthwhile if you provide such deeper reflection in the discussion section beyond what you now state on page 18: “Clearly, if the primary aim of an intervention is to increase healthy behaviour, completely replacing regular products with healthy products has the greatest inpact”.

2. As your data are at vending machine level, it is clear that you do not have access to individual level purchases. But one rival explanation could be that “snack preferences are heterogenous across population”, “people choose according to their product preferences”, and “different vending machine assortments attract different consumer segments”. Essentially your paper would then indicate that the segment of “calorie-lovers” is larger than that of “health motivated consumers”. Maybe this is an explanation you could also discuss as your results also show that these impulses are not strong enough for individual to make an effort to obtain the product from elsewhere if it is not available in the vending machine. Related to this point, you do unfortunately not report on the price levels of the items (only the margin which is surprisingly similar across healthy and unhealthy items?). But, another alternative explanation could be that higher priced assortments (which I assume the healthy options will be) appeal to a different (and smaller) consumer segment. Could you please reflect on the prices and reveal them in the table in supplementary materials?

3. Also I think that the discrepancy between significantly fewer purchases, but non-significant loss in profit (despite the fact that average margin is similar (42 p) across the two conditions, is not very convincing. It is not fully clear to me how you relate this to “heterogeneity” as an explanation. That could use some more discussion, also in managerial terms as I assume that the relevant commercial benefit for the hospital is at vending machine level, not item level?

4. The other issue is the external validity of your findings that could use some further elaboration. With a 30% drop in profit and the large across item variability in sales (both for healthy and unhealthy category by the way) illustrated in your figure 5 what would be key managerial implications for vending machine managers? Could you reflect on whether such “all healthy” business model could be attainable for them? This could use more in depth discussion in the discussion section.

Reviewer #2: This is a well-written paper which reports a very valuable and carefully conducted field experiment on the effect of removing unhealthy products from vending machines in a hospital setting.

This type of field research is very much needed, as it seems quite extreme to remove all unhealthy items from a vending machine, but the study shows that it is possible. Moreover, what it particularly useful is that the authors examined compensatory behaviors in different ways, which is real risk when these interventions are implemented. The literature review is solid and the authors carefully discuss their findings and study limitations in the discussion section. One concern that is have is related to the nutritional guidelines to determine whether a product is relatively healthier or not. These criteria are 'best in product class' criteria, which means that they might be not that strict. It would be helpful to provide more information and examples of products that do and do not meet the criteria. Are the Welsh Hospital Healthy Vending directive constraints developed in cooperation with the food or catering industry? How are they different from other nutritional standards such as those developed for schools?

6. PLOS authors have the option to publish the peer review history of their article (what does this mean?). If published, this will include your full peer review and any attached files.

Reviewer #1: No

Reviewer #2: No

---

## [Author Response · Author response to Decision Letter 0]

24 Jul 2020

Changes made in response to reviewer comments

Reviewer 1

Comment: 

“1. What do the results actually show us? In other words what is the substantive contribution to the field?

Does not the paper essentially show that if (mildly) hungry people face an assortment they will select an option that best fits their preference structure (or refrain from buying if it is not there). This, by definition, will lead to fewer and lower caloric choices in the healthy assortment condition. This is because, assuming that preferences are, on average, biased towards less healthy options, it is less likely that a preferred option is available and if it is, in the healthy assortment, it will by definition lead to reduced caloric intake.

It would be worthwhile if you provide such deeper reflection in the discussion section beyond what you now state on page 18: “Clearly, if the primary aim of an intervention is to increase healthy behaviour, completely replacing regular products with healthy products has the greatest impact”.”

Response: 

We have altered the manuscript to make the primary contribution clearer. We argue that complete healthy vending does not by definition lead to lower caloric consumption overall because individuals may have engaged in compensatory behaviors to make up for the lower calorie count of the healthy products. Our data demonstrates that this did not happen. Furthermore, while sales were lower in the healthy condition (although not significantly so), the drop was surprisingly small. Consumers seem relatively flexible in what they will purchase from a vending machine: if their regular products are not present, they simply purchase something else, even if it is a healthy product. 

Comment: 

“2. As your data are at vending machine level, it is clear that you do not have access to individual level purchases. But one rival explanation could be that “snack preferences are heterogenous across population”, “people choose according to their product preferences”, and “different vending machine assortments attract different consumer segments”. Essentially your paper would then indicate that the segment of “calorie-lovers” is larger than that of “health motivated consumers”. Maybe this is an explanation you could also discuss as your results also show that these impulses are not strong enough for individual to make an effort to obtain the product from elsewhere if it is not available in the vending machine. Related to this point, you do unfortunately not report on the price levels of the items (only the margin which is surprisingly similar across healthy and unhealthy items?). But, another alternative explanation could be that higher priced assortments (which I assume the healthy options will be) appeal to a different (and smaller) consumer segment. Could you please reflect on the prices and reveal them in the table in supplementary materials?”

Response: 

This is a very interesting idea. We have added a paragraph discussing it in the GD. 

We agree that different priced products will appeal to different consumer segments, and to control for this, we priced all products at 80p. This was originally noted in the “Vending snacks” section of the methods. We have now further emphasized the rationale behind our pricing in this section, and have added product cost and profitability information for all products in S4 Appendix. 

Comment: 

“3. Also I think that the discrepancy between significantly fewer purchases, but non-significant loss in profit (despite the fact that average margin is similar (42 p) across the two conditions, is not very convincing. It is not fully clear to me how you relate this to “heterogeneity” as an explanation. That could use some more discussion, also in managerial terms as I assume that the relevant commercial benefit for the hospital is at vending machine level, not item level?”

Response: 

We found the same pattern of results for sales volume and profit, contrary to the reviewer’s comment above. We have now made this clearer in the discussion section. 

The commercial benefit for the hospital is at the overall profit level, not the item level, but we are not sure why the reviewer has this concern.

Comment: 

“4. The other issue is the external validity of your findings that could use some further elaboration. With a 30% drop in profit and the large across item variability in sales (both for healthy and unhealthy category by the way) illustrated in your figure 5 what would be key managerial implications for vending machine managers? Could you reflect on whether such “all healthy” business model could be attainable for them? This could use more in depth discussion in the discussion section.”

Response: 

We have added a new section, “Implications for vendors”, that helps address this query. 

Reviewer 2

Comment: 

“One concern that I have is related to the nutritional guidelines to determine whether a product is relatively healthier or not. These criteria are 'best in product class' criteria, which means that they might be not that strict. It would be helpful to provide more information and examples of products that do and do not meet the criteria. Are the Welsh Hospital Healthy Vending directive constraints developed in cooperation with the food or catering industry? How are they different from other nutritional standards such as those developed for schools?”

Response: 

We have altered the manuscript to provide more clarity on the nutrients targeted in the guidelines, and have highlighted that they were developed by the Welsh government. More details are now accessible in S5 Appendix, which lists the maximum level of fat, saturated fat, added sugars and salt allowed in any product, and any exemptions that may apply. We have also made reference to the generalizability of the results to school settings which already have their own, arguably stricter, guidelines in place.

---

## [Decision Letter · Decision Letter 1]

8 Sep 2020

The health benefits and cost-effectiveness of complete healthy vending

PONE-D-20-11945R1

Dear Dr. Griffiths,

We’re pleased to inform you that your manuscript has been judged scientifically suitable for publication and will be formally accepted for publication once it meets all outstanding technical requirements.

Kind regards,

Zhifeng Gao

Academic Editor

PLOS ONE

Additional Editor Comments (optional):

Reviewers' comments:

Reviewer's Responses to Questions

**Comments to the Author**

1. If the authors have adequately addressed your comments raised in a previous round of review and you feel that this manuscript is now acceptable for publication, you may indicate that here to bypass the “Comments to the Author” section, enter your conflict of interest statement in the “Confidential to Editor” section, and submit your "Accept" recommendation.

Reviewer #1: All comments have been addressed

Reviewer #2: All comments have been addressed

2. Is the manuscript technically sound, and do the data support the conclusions?

Reviewer #1: Yes

Reviewer #2: (No Response)

3. Has the statistical analysis been performed appropriately and rigorously? 

Reviewer #1: Yes

Reviewer #2: (No Response)

4. Have the authors made all data underlying the findings in their manuscript fully available?

Reviewer #1: Yes

Reviewer #2: (No Response)

5. Is the manuscript presented in an intelligible fashion and written in standard English?

Reviewer #1: Yes

Reviewer #2: (No Response)

6. Review Comments to the Author

Reviewer #1: No further comments for the authors. Satisfied by their response and how it was incorporated within the revised manuscript

Reviewer #2: (No Response)

7. PLOS authors have the option to publish the peer review history of their article (what does this mean?). If published, this will include your full peer review and any attached files.

Reviewer #1: No

Reviewer #2: No

---

## [Editor Report · Acceptance letter]

11 Sep 2020

PONE-D-20-11945R1 

The health benefits and cost-effectiveness of complete healthy vending 

Dear Dr. Griffiths:

I'm pleased to inform you that your manuscript has been deemed suitable for publication in PLOS ONE. Congratulations! Your manuscript is now with our production department. 

Kind regards, 

on behalf of

Dr. Zhifeng Gao 

Academic Editor

PLOS ONE